# Effects of Mixed Pasture Legume Phytoestrogens on Superovulatory Response and Embryo Quality in Angus Cows

**DOI:** 10.3390/ani14071125

**Published:** 2024-04-07

**Authors:** Jessica M. Wyse, Rory P. Nevard, Jaymie Loy, Paul A. Weston, Saliya Gurusinghe, Jeffrey McCormick, Leslie A. Weston, Cyril P. Stephen

**Affiliations:** 1School of Agricultural, Environmental and Veterinary Science, Charles Sturt University, Wagga Wagga, NSW 2650, Australia; jloy@csu.edu.au (J.L.);; 2Gulbali Institute for Agriculture, Water and the Environment, Charles Sturt University, Wagga Wagga, NSW 2650, Australiapweston@csu.edu.au (P.A.W.); sgurusinghe@csu.edu.au (S.G.); leweston@csu.edu.au (L.A.W.); 3Sydney School of Veterinary Science, The University of Sydney, Camden, NSW 2570, Australia

**Keywords:** lucerne, phytoestrogens, superovulation, follicle, corpus luteum, embryo

## Abstract

**Simple Summary:**

Phytoestrogens are secondary plant metabolites that are structurally similar to endogenous estrogens and may result in adverse reproductive effects in grazing livestock. This study evaluated the key phytoestrogens in legume pastures and their effect on superovulation, embryo quantity and embryo quality. Cows grazed on legume and ryegrass pastures for a duration of 7 weeks at two timed treatments and were subjected to a conventional estrus synchronization and superovulation protocol. Coumestrol and formononetin were identified as the dominant phytoestrogens. The results from the grazing trial suggest that grazing potential estrogenic legume pastures less than 20 days before artificial insemination may affect oocyte developmental competence and contribute to early embryonic loss.

**Abstract:**

Ovulation and artificial insemination rates have been observed to decrease in sheep and cows when exposed to dietary phytoestrogens at concentrations greater than 25 mg/kg DM. A grazing trial was undertaken to investigate the effects of coumestrol and other key phytoestrogens on the superovulatory response, embryo numbers and quality in beef cows grazing legume pastures. A 7-week controlled grazing trial was conducted with legume and ryegrass pasture treatments, with cows exposed to legumes at two timed treatments, 4 and 7 weeks. Twenty Angus cows were subjected to a conventional estrus synchronization and superovulation protocol. Embryos were recovered via conventional uterine body flushing 7 days post artificial insemination (AI). Numerous phytoestrogens were identified in both pasture and plasma samples, including coumestrol and formononetin. Concentrations of phytoestrogens in the pasture ranged from 0.001 to 47.5 mg/kg DM and 0 to 2.6 ng/mL in plasma. Approximately 50% of cows produced viable embryos 7 days post AI. A significant interaction between the effect of treatment groups on the embryo stage was observed (*p* < 0.05). The results suggest that concentrations of >25 mg/kg DM of phytoestrogens less than 20 days preceding AI may negatively affect oocyte developmental competence, reduce progesterone production and thus contribute to early embryonic loss.

## 1. Introduction

Pasture legumes in Australia remain important feed and fodder sources for grazing livestock, with newer cultivars continually developed for our unique agricultural systems. The perennial legume, lucerne (*Medicago sativa*), is widely grown and adapted to a diverse range of environments, while white clover (*Trifolium repens* L.) is predominantly used in high-rainfall and irrigation districts [1]. Pasture legumes are highly valued in the mixed farming systems of southern Australia. In combination with the use of superphosphate, they have led to successful livestock production [2,3] and increased crop yields when grown in rotation [1]. The addition of pasture legumes into perennial pastures has also improved overall nutritional quality by increasing the availability of digestible nutrients and protein for grazing livestock [4,5]. Lucerne, in particular, is widely utilized in mixed farming systems [6] and can be applied as a forage during the mating season as it provides a rising plane of nutrition for grazing livestock. Of all common pasture legumes, lucerne provides a higher concentration of protein and minerals for livestock and a lower fiber content, making it beneficial for the provision of higher net energy [7]. Lucerne has a higher voluntary intake by ruminants, compared to that of grasses, due to the higher availability of dry matter in the form of cell solubles [8]. These are readily available and can be easily absorbed in the digestive tract. [8]. Lucerne also plays an important role in nitrogen fixation and the utilization of sub soil moisture, reducing water table recharge and dryland salinity [9]. However, despite these benefits, pasture legumes typically produce bioactive secondary plant metabolites that have been identified to cause reproductive problems in grazing livestock along with bloating and a reduced palatability of forages [6,10,11,12].

Phytoestrogens are defined as non-essential secondary plant metabolites that are positively associated with plant defense [13]. Estrogenic *Medicago* spp. and *Trifolium* spp. pastures, which produce significant levels of phytoestrogens (>25 mg/kg DM [14]), have been found to affect fertility in cattle, sheep and horses, mainly in years where pastures have been affected by pest infestations or other biotic or environmental stressors. Phytoestrogens and their metabolites specifically decrease the release of luteinizing hormone (LH) and inhibit the response of gonadotropins on granulosa cells [15] by affecting the hypothalamic–pituitary–gonadal (HPG) axis and interfering with the endogenous feedback of 17β-estradiol [16]. They can also indirectly increase prostaglandin F2α (PGF2α) secretion from the endometrium in cows and affect the development and maturation of ovarian follicles [15]. Livestock grazing pastures with high phytoestrogen concentrations may also experience reduced reproductive efficiency characterized by lower conception rates and increased embryonic loss [16].

It has been observed that cows are more sensitive than sheep to the influence of isoflavones [17] on their estrous cycles. This is likely due to the isoflavone inhibition of aromatase, which is essential for estrogen biosynthesis and the development of ovarian follicles [17]. The suppression of follicular development through this action has been observed, along with the stimulation of multi-oocyte production [18]. Currently, genistein is the only isoflavone known to block oocyte growth and disrupt follicle morphology [19,20]. Coumestrol, a type of coumestan, has been observed to impair the secretory function of the corpus luteum (CL) by increasing oxytocin and decreasing prostaglandin E2 (PGE2) in luteal cells. Early embryonic loss has been noted to be as great as 29% in cows when fed on a legume pasture, compared to only 14% when fed on a non-legume pasture [21]. Exposure to multiple phytoestrogenic metabolites during key periods of fertilization and development may therefore have the capacity to alter the function of the bovine reproductive system. 

Coumestrol has been identified in several legume pasture species and has been recognized as one of the most potent phytoestrogens for the inhibition of estrus and resulting ovulatory failure [22]. Recent advances in analytical chemistry have enabled the rapid and accurate profiling of these key secondary metabolites with enhanced sensitivity and detection at parts per billion (ppb) or trillion (ppt) levels [23]. Therefore, the objectives of this research were specifically to (a) develop a highly sensitive method to accurately characterize metabolites in both lucerne pasture samples and bovine plasma using liquid chromatography coupled with mass spectrometry (LC/MS QToF) to perform non-targeted metabolite profiling, and then perform a quantification of key metabolites using liquid chromatography coupled with triple quadrupole mass spectrometry (LC/MS QQQ), building on the previous method established by Wyse et al. [6]; (b) characterize and quantify the coumestans and the other relevant phytoestrogens detected in bovine plasma and evaluate their influence on follicular development and embryonic development at 7 days post AI in grazing livestock; and (c) to determine at what concentrations the superovulatory response and embryo quality in beef cows grazing pasture legumes are affected by phytoestrogens.

## 2. Materials and Methods

The use of cattle in this experiment was approved by the Animal Care and Ethics Committee, Charles Sturt University—Protocol No. A21368.

### 2.1. Site Information

All experimentation was performed at the Charles Sturt University Cattle Farm, Wagga Wagga, New South Wales. This trial was conducted during the late spring/early summer of 2021 using a selection of pasture types to evaluate the impact on reproductive fecundity in grazing cows. The botanical composition of specific pasture treatments is presented in Table 1. The treatment and control groups were as follows: L7 (seven weeks on lucerne), L4 (three weeks on an annual ryegrass (*Lolium multiflorum*) pasture, followed by four weeks on lucerne), GC7 (seven weeks on a ryegrass/clover mix pasture), and control (G7—seven weeks on ryegrass pasture). The GC7 treatment was similar in the composition of the annual to the control (G7) but contained a significantly higher percentage of legume species, compared to the G7 group itself, which contained isolated occurrences of naturalized clover species not understood to be estrogenic (e.g., haresfoot clover). Pasture composition was assessed visually in each of the four grazing treatments utilizing 35 × 35 cm^2^ quadrants across a diagonal transect of the field plots. Rainfall and temperature data were recorded for the duration of the experiment at Wagga Wagga Bureau Station, Wagga Wagga, 7.8 km away from the field site.

The clover species identified in the treatments were identified as a ‘wild’ type as they were not sown into the mixed pasture. The percentages of the other legumes present in the GC7 treatment were as follows: cluster clover 10%, haresfoot clover 4%, arrowleaf clover 3%, white clover 1% and barrel medic 1%. Arrowleaf clover, white clover and barrel medic have been previously identified to contain phytoestrogens. The two naturalized clovers are currently not reported to contain any of the key phytoestrogens being investigated in this trial.

### 2.2. Study Design, Feeding and Animals

Twenty pluriparous Angus cows between 52 and 96 days post-partum and with a body condition score (BCS) range of 3–4 were selected. Cows ranged in age from eight to ten years. Before introduction to the trial, the cows were grazing on a mixed grass pasture. On day 0 (trial commencement), cows were randomly allocated into one of four experimental groups (*n* = 5 per group) in a randomized incomplete block design. Pasture lucerne consisted of a 6-year-old stand of the cultivar, ‘Genesis’. Genesis is a dual-purpose winter active lucerne cultivar (winter activity rating 7). It is resistant to pests such as the spotted alfalfa aphid (*Therioaphis trifolii*) and blue green aphid (*Acyrthosiphon kondoi*) and diseases such as phytophthora root rot (*Phytophthora medicaginis*) and collectotrichum crown rot (*Colletotrichum trifoii*). It is suited to most soil types. As a highly productive cultivar, it is extremely adaptable for both grazing and hay production and as a ley legume in cropping rotations. Cows had access to ad libitum water. As a preventative measure, 25 mL of Vicchem Oral Bloat Drench^TM^ (active ingredient: C12–C15 ethoxylated alcohol) was administered to each individual on day 0 of the trial and was also introduced to the water troughs at a ratio of 0.26:1 L (drench/water). The BCS was maintained at 4 (out of 5) for the duration of the trial.

### 2.3. Synchronization of Estrus and Superovulation

Cows were subjected to a conventional estrus synchronization and superovulatory protocol (Figure 1) similar to those described by Mapletoft and Bo (2014) [24]. This consisted of a combination of progesterone, estradiol, prostaglandinF2α and gonadotropin-releasing hormone (GnRH). Intravaginal progesterone-releasing devices (Eazi-Breed Cattle CIDR^®^, Zoetis Inc., Parsippany, NJ, USA) containing 1.38 g of progesterone per CIDR^®^ were administered on day 1. An intramuscular injection of 2 mL Estrumate^®^ (containing 250 μg/mL of cloprostenol as the sodium salt, Merck Animal Health, Rahway, NJ, USA) was administered to each cow 10 days prior to the insertion of the CIDR^®^ devices. A week after the injection of Estrumate^®^, ovaries of all individuals were scanned by ultrasound (Logiq V2 Ultrasound, GE Healthcare, Chicago, IL, USA) for the presence of corpus luteum (CL). On the day of the CIDR^®^/IVD insertion, individuals received a 2 mL intramuscular injection of Bomerol^TM^ (containing 1 mg of estradiol benzoate/mL, Bayer Australia Ltd., Pymble, Australia). Ovaries were then scanned (Logiq V2 Ultrasound, GE Healthcare, USA) 4 days later to assess the number of follicles, size of the largest follicle and number of CLs. Folltropin^®^-V (containing 400 mg NIH-FSH-P1, Vetoquinol, Hamilton, Australia) was injected twice daily over a period of 4 days (on day 4, 5, 6 and 7 post-CIDR^®^ insertion) in a tapering dose (2.4 mL, 2 mL, 1.6 mL, 1.2 mL) to stimulate the ovaries for superovulation. On the last day of Folltropin^®^-V injection, an intramuscular injection of Estrumate^®^ (2 mL) was given in the morning and afternoon, following which the CIDR^®^ was removed, and the cows were marked with tail paint to assist in estrus detection. An intramuscular injection of 2.5 mL Cattle-Mate^TM^ (containing 100 μg gonadorelin/mL, Vetoquinol, Hamilton, Australia) was administered the following day, and ovaries were scanned for the number of follicles and size of the largest follicle. The cows were artificially inseminated using frozen semen the following day. All cows underwent a transrectal ultrasound scan (Logiq V2 Ultrasound, GE Healthcare, Chicago, IL, USA) to confirm the presence of CLs two days after insemination. All procedures were carried out by qualified veterinary practitioners.

### 2.4. Blood Sampling

To determine the plasma concentrations of key coumestans and isoflavones, 10 mL of blood was collected from the coccygeal vein using BD Vacutainer^®^ Heparin tubes (containing lithium heparin or sodium heparin as an anticoagulant) and a 20 g × 1″ needle (PrecisionGlide^TM^, Becton Dickinson, Mulgrave, Australia). Tubes were gently inverted to ensure the dispersal of the anticoagulant. Samples were kept chilled before undergoing centrifugation (Eppendorf Centrifuge 5810, 4000 rpm for 10 min; Eppendorf, Macquarie Park, Australia) to extract the plasma fraction. Plasma samples were stored at −20 °C until further processing for the extraction of phytoestrogens. Samples were collected from the cows weekly, with the initial samples collected prior to the commencement for the estimation of baseline concentrations (Table 2).

### 2.5. Embryo Recovery and Assessment of Quality

Embryos were recovered non-surgically via conventional uterine body flushing, similar to a previously described method [25] by two veterinarians on day 7 post AI. Approximately 1.5–2 L of flush media (BoviFlush embryo recovery medium, Minitube) was used per donor depending on the capacity of the uterus. The recovered flush fluid was filtered (EmSafe embryo filter, Minitube) and transferred into a Petri dish (35 mm, Minitube) and examined under a stereomicroscope (Leica DM IL LED Inverted Microscope, Leica Microsystems, Germany) to identify the number, grade and quality of the embryos. Following this, the embryos were transferred into a 5-well embryo dish (5-well dish for embryo culture, Minitube), and wells were labeled with the specific cow’s ID number. The grading of the embryos was based on the International Embryo Transfer Society (IETS) [26], on a 1–8 stage code (1. Unfertilized oocyte; 2. One-celled embryo; 3. Morula; 4. Compact morula; 5. Early blastocyst; 6. Blastocyst; 7. Expanded blastocyst; 8. Hatched blastocyst) and a 1–4 quality code (1. Excellent; 2. Fair; 3. Poor; 4. Dead or degenerating). The evaluation of the embryos was performed by a non-objective technician using blind evaluations of the embryos.

### 2.6. Pasture Sampling

Pastures were sampled and the botanical composition was recorded weekly according to the modified method by McIntyre (1952) [27] using a 35 × 35 cm^2^ quadrat (Figure 2a). Six composite samples were taken across a diagonal transect of the individual pasture treatments spaced approximately 50 m apart (Figure 2b). A composite sample weighing approximately 300 g was taken [6]. Three composite samples of the arrowleaf clover, cluster clover and haresfoot clover were also collected at three time points (1, 3 and 7 weeks) to analyze the phytoestrogen content in these clovers and their contribution to the GC7 treatment (Appendix B, Table A1). Samples were kept chilled, before being stored at −20 °C until extraction and further analysis by high performance liquid chromatography coupled with quadrupole time-of-flight mass spectrometry (HPLC-MS-QToF) and liquid chromatography coupled with triple quadrupole mass spectrometry (LC-QQQ-MS/MS). The samples were defrosted at room temperature, and 5 g of each of the composite tissue samples was utilized for extraction.

Pasture height was recorded over a diagonal transect across the treatment pastures at 50 sites for each treatment (Figure 2b) using modified methods by Rayburn et al. (Year) [28] outlined in the PROGRAZE Manual [29]. The height of the pasture was recorded for the green-growing species. The pasture sample’s measurement number was multiplied by the number of pasture heights; the total height of the pasture was then averaged over the number of measurements. The corresponding estimated pasture density values were assessed to estimated pasture mass (kg DM/ha). This was recorded weekly for seven weeks. The estimated pasture masses for the treatments were as follows: L7 and L4, 3300–3000 kg DM/ha; GC7, 2500–2000 kg DM/ha; and G7, 2000–1700 kg DM/ha over the duration of the trial.

### 2.7. Analytical Method

Key plant-produced metabolites (coumestrol and the isoflavones biochanin A, daidzein, formononetin and genistein) were quantified in pasture and plasma samples via UHPLC-MS-QQQ. In addition, pasture samples were analyzed via UHPLC-MS-QToF in order to assess the abundance of other phytoestrogens for which analytical standards were not available. This permitted the estimation of the relative abundance of the compounds among treatment groups and later in plasma samples even though the absolute concentration could not be determined.

Plant material and plasma were extracted using the method developed by Wyse et al. (2021) [7]. The metabolomic profiling of lucerne samples was performed similarly to Wyse et al. (2021) using an Agilent 1290 Infinity UHPLC and an Agilent 6530 Quadrupole Time-of-Flight (QToF) Mass Spectrometer (MS) (Agilent Technologies, Santa Clara, CA, USA). Full scan mass spectra were measured over a *m/z* range of 50–1700 Da at a rate of two spectra/second in the negative ion mode. Chromatographic separation was carried out using a reverse-phase Poroshell 120 SB-C18 column (2.1 × 100 mm, 2.7 µm particle size) (Agilent Technologies, Santa Clara, CA, USA) equipped with a guard column (Poroshell 120 Bonus-RP, 2.1 mm, 2.7 um particle size) (Agilent Technologies, Santa Clara, CA, USA) using a flow rate of 0.3 mL min^−1^. The column was equilibrated for 30 min prior to analysis. Separation was obtained using a gradient of solvent A [water (Milli-Q, TKA-GenPure, TKA Thermo Scientific, Waltham, MA, USA) + 0.1% formic acid (LC-MS grade, LiChropur^®^, 98–100%, Sigma-Aldrich, St. Louis, MO, USA)] and solvent B [95% HPLC-grade acetonitrile (RCI Labscan, Bangkok, Thailand) + 0.1% formic acid]. The solvent gradient was as follows: 10% B over 0–1 min, increasing to 80% B from 1 to 11 min, then increasing further to 100% B over 11–13 min and holding there until 17 min. The solvent composition returned to 10% B from 17 to 18 min and held there until 24 min. The injection volume for each sample was 10 µL. Nitrogen was used as the drying gas at 250 °C at a flow rate of 9 L min^−1^. The phytoestrogens identified in the lucerne samples are listed in Table 3.

The absolute quantitation of key phytoestrogens (coumestrol and isoflavones) in pasture and plasma samples was performed utilizing dynamic multiple reaction monitoring (dMRM) with an Agilent 6470 Triple Quadrupole UHPLC/MS. Chromatographic separation was carried out as described above for UHPLC-MS QToF analyses but with a modification of the solvent gradient as follows: 25% B over 0–1 min, then increasing to 75% B from 1 to 12 min, then increasing further to 100% B over 0.1 min and holding there until 15 min. The solvent composition returned to 25% B from 15 to 15.1 min and held there until 22 min. The injection volume for each sample was 5.0 µL. Nitrogen was used as the drying gas at 250 °C at a flow rate of 9 L min^−1^. Metabolites were identified by the match to accurate mass and retention times (RTs) of analytical standards (STD) and the match to accurate mass and molecular formula (AM) using a personal compound database library (PCDL). The phytoestrogens identified in samples and the dMRM parameters used for each are listed in Appendix A. Quantification was performed by external calibration, and the results are expressed in mg/kg of dry matter (DM) using DM% obtained from drying pasture samples. Analyte standards were prepared, and concentration curves were obtained over the tested concentration range (0.001–10 ug/mL) with correlation coefficients *R* ≥ 0.9651 for each of the phytoestrogens. The concentrations of the replicates were averaged per sampling date.

## 3. Results

### 3.1. Quantification of Phytoestrogens in Pasture

A range of phytoestrogens were identified in methanolic extracts of pasture legumes and grass species by HPLC-QTOF-MS in negative ionization mode. These included 3′methoxycoumestrol, 4′methoxycoumestrol, coumestrol, biochanin A, daidzein, formononetin, genistein and quercetin. 3′ and 4′methoxycoumestrol were annotated by their respective molecular features including retention time, accurate mass and molecular formulae along with several glyco- and glucosides, apigenin-7-*O*-glycoside, luteolin-7-*O*-glucoside, coumestrol-3-*O*-glucoside and isorhamnetin 3-O-glucoside.

The concentrations of phytoestrogens identified by internal standards ranged from 0 to 47.5 mg/kg DM: biochanin A (0.003–4.3 mg/kg DM), coumestrol (0–41.2 mg/kg DM), daidzein (0–1.2 mg/kg DM), formononetin (0.01–12.5 mg/kg DM) and genistein (0.001–7.2 mg/kg DM). The concentrations of phytoestrogens quantified in G7 remained under 5 mg/kg DM for the duration of the grazing experiment (Figure 3a). The profiles of the phytoestrogens in both lucerne treatments L4 (Figure 3b) and L7 (Figure 3c) were similar, with coumestrol and formononetin being the dominant phytoestrogens quantified. Coumestrol was present in high concentrations on sampling date 28 for L4 and L7 treatments, at concentrations of 41.2 mg/kg DM and 24.2 mg/kg DM, respectively. These concentrations exceed the suggested safety threshold for coumestrol of 18 mg/kg DM [14]. For the GC7 treatment, coumestrol and the isoflavones (biochanin A, daidzein, formononetin, genistein) were detected at low concentrations across all eight of the sampling dates (Figure 3d). There was a significant difference observed for the concentration of the phytoestrogens between the treatments (*p* < 0.05, LSD: 8.92). Formononetin was consistently detected in greater concentrations compared to the other phytoestrogens present in the GC7 pasture, except for the last sampling date. When compared to the concentrations of phytoestrogens in G7, the 20% of legumes present in the mixed pasture (GC7) contributed to an increase in phytoestrogen concentrations in these samples compared to the ryegrass in the control (Appendix B, Table A1). The concentrations of all individual and total phytoestrogens quantified remained under the recommended safety threshold level of 25 mg/kg for the duration of the grazing trial, except for day 28 in the L4 treatment. Of the coumestans, coumestrol was found in the highest relative abundance compared to the other four coumestans tentatively identified in the pasture samples (Appendix B, Figure A1). Formononetin was also found in greatest abundance compared to the other isoflavones.

### 3.2. Quantification of Phytoestrogens in Plasma of Grazing Cows

A range of phytoestrogens was also identified in bovine plasma by HPLC-QQQ-MS in negative ionization mode, including coumestrol, biochanin A, daidzein, formononetin and genistein. 4′methoxycoumestrol and 3′methoxycoumestrol were annotated and identified by accurate mass and molecular formula. The concentrations of coumestrol between the treatment groups ranged from 0.16 to 2.6 ng/mL, with the highest recorded in the initial sampling in the GC7, L7 and L4 treatment groups. The concentration differences between the treatment groups were not observed to be significant (*p* > 0.05, LSD: 8.68). However, the sampling date was observed to have a significant impact on coumestrol, biochanin A, formononetin and genistein concentrations (*p* < 0.01) in the plasma (Figure 4). The sampling date also significantly affected daidzein concentrations (*p* < 0.05) (Figure 4). Concentrations of biochanin A decreased for both the treatment groups and the control over the sampling dates (Figure 4a). Concentrations of phytoestrogens decreased in plasma for the G7 group from 0.16 ng/mL to 0.0042 ng/mL.

### 3.3. Follicular Development

Ovaries were scanned for the number and size of follicles at day 35 (4 days after CIDR^®^ insertion) or day 39 (1 day prior to AI) (Table 4) using Logiq V2 Ultrasound (GE Healthcare, Chicago, IL, USA) (Figure 5). No significant differences were observed for the average number of follicles of any size between the treatment groups on either day 35 or day 39. There was a significant difference in the number of follicles > 1 cm between day 35 and day 39 (*p* < 0.05) and a significant difference in the number of follicles < 1 cm between the two days (*p* < 0.05).

The GC7 group had the greatest number of follicles < 1 cm, and the L7 group had the greatest number of follicles > 1 cm (Table 4). There was a significant difference in the average number of antral follicles between day 35 and day 39 (*p* < 0.05). Antral follicles were described as having a fluid-filled antrum, also known as tertiary follicles [30], that were visible on the ultrasound, irrespective of their size. The greatest versus least numbers were based on the number of antral follicles seen on the ultrasound. The L7 group also had the greatest number of antral follicles, while the L4 group had the fewest antral follicles. The G7 and GC7 groups had the same number of antral follicles on both day 35 and day 39.

### 3.4. Embryo Quantity and Quality

A total of 56 embryos and 8 unfertilized oocytes (UFOs) were recovered from the embryo flush of 19 out of the 20 cows, seven days after AI. Embryos were retrieved from 10 of the 19 cows (75% of the cows in G7, 40% in L4, 100% in L7 and 0% in GC7). Overall, 47% of cows produced embryos 7 days post AI. The quality of the embryos ranged from grades 1 to 3 in the G7 and L7 groups (Table 5). All cows from the L4 group produced grade 1 embryos; however, none were recovered from the GC7 group. There were no differences in the embryo quality between the replicates for the treatment groups L7 and L4 (*p* > 0.05). A two-way ANOVA revealed the effect of the treatment groups on embryo quality was not significant (F(9,64) = 1.08, *p* > 0.05).

Most of the embryos recovered were of a similar stage; 4 (compact morula), 5 (early blastocyst) and 6 (blastocyst) (Figure 6). The embryo stage differed between the replicates within the individuals in G7 (*p* < 0.05) and L7 (*p* < 0.01), but no significant difference in the embryo stage between the replicates for treatment group L4 was observed (*p* > 0.05). A two-way ANOVA performed to analyze the effect of the treatment group on the embryo stage revealed an interaction between the effect of the treatment groups and the embryo stage (F(7,21) = 2.34, *p* < 0.05).

## 4. Discussion

This field study provides evidence that grazing cows on a predominantly lucerne pasture (>70% by area) which possesses high concentrations of phytoestrogens (>25 mg/kg DM) 20 days prior to artificial insemination may have a negative effect on embryo grade and thereby impact pregnancy outcomes. Furthermore, grazing on pastures with >70% lucerne with low concentrations of phytoestrogens for a longer duration may reduce the risk of detrimental reproductive effects, as the rumen may have had sufficient time to adapt to the presence of phytoestrogens [18,31]. The phytoestrogens, genistein, daidzein and coumestrol, have been observed to impact several processes during early conception. During days 11–15 of the estrous cycle, coumestrol has been observed to impair the secretory function of the CL by increasing oxytocin and decreasing PGE2 in luteal cells [15]. Mlynarczuk et al. (2011) [15] suggested that coumestrol resulted in persistent CLs, the impairment of ovulation and the formation of ovarian cysts in cows. In the current study, the formation of persistent CLs may be prolonging the inter-estrus interval rather than directly impairing ovulation and thus was reducing reproductive efficiency. As recent research on the impact of phytoestrogens on fertilization and embryo development in vivo was conducted in mice and sheep, our study is one of the very few investigating the in vivo effects of phytoestrogens on follicular development, superovulation, fertilization and embryo development in cows.

### 4.1. Metabolomic Profiling of Key Phytoestrogens in Legume Pastures and Bovine Plasma

The dominant phytoestrogens recovered in the pasture samples were coumestrol and formononetin. Whilst there was a week-long exposure to a blue green aphid infestation observed in treatments L4 and L7, it did not appear to affect the chemical profile and production of the phytoestrogens in these treatments. The concentrations of coumestrol observed in the pasture during the trial exceeded 40 mg/kg DM in the L4 pasture on day 28, 12 days prior to AI. The combined accumulation, however, of the five quantified phytoestrogens only exceeded the safety threshold of 25 mg/kg DM on day 28 in the L7 pasture. The combined accumulation of phytoestrogen concentrations peaked at 47.5 mg/kg DM in the L4 pasture and 30.8 mg/kg DM in the L7 pasture on day 28, 12 days prior to AI. The coumestans, 3′methoyxcoumestrol, 4′methoxycoumestrol and coumestrin (coumestrol 3-*O*-glucoside), were annotated by their respective molecular features. In terms of relative abundance, 3′methoxycoumestrol was present in a higher relative abundance compared to coumestrol (Appendix B Table A1). The negative impacts of 3′methoxycoumestrol, 4′methoxycoumestrol and coumestrin upon bovine reproduction are currently unexplored, but it is likely that methoxycoumestrols undergo demethylation to coumestrol [4].

Of the phytoestrogens quantified, coumestrol was present in the greatest concentration in the plasma, at 0.3 ng/mL during the grazing trial. This is lower than the concentrations recorded in plasma in a prior grazing trial, where concentrations of coumestrol were recorded at 2.6 ng/mL [6]. The concentrations of coumestrol reported in this study were lower than previously observed in a pure stand of lucerne *cv.* Genesis [6]. One factor attributed to this could be due to the dilution of L4 and L7 by the annual ryegrass. The resulting coumestrol concentrations analyzed in plasma were also lower than the previous study conducted by our group [6] but are higher than other concentrations reported in bovine plasma, 0.1 ng/mL [32]. The concentrations of biochanin A in the plasma of the GC7 treatment remained higher than the other treatment groups for the duration of the grazing trial. This may be attributed to the presence of clovers in the GC7 treatment. While sampling in GC7 accounted for 19% of the clover in the collected pasture samples, the voluntary grazing intake of the clover may be higher in the cows, contributing to concentrations of biochanin A and formononetin in the plasma.

Of the 20 cows that underwent AI, only 47% had a positive flush for embryos on day 7 post AI. The embryo recovery efficiency is quite low, compared to the industry standard of 8–10 total embryos per donor [33]. Only three cows in the G7 group and one cow in the L4 group were within the industry standard outlined for embryo recovery. High concentrations of phytoestrogens in pastures have been previously demonstrated to alter conception rates in grazing cows, especially when aphid or pathogen infestations are present. Coumestrol concentrations of silage above 66.8 mg/kg of DM achieve 376 pregnancies out of 1264 inseminations performed, i.e., only 30.17% [34]. It has also been noted that the fecundity of the flock or herd may influence the effect observed, with sudden lower conception rates more noticeable in herds that generally would have high conception rates. [35].

The relative potency of phytoestrogens in ruminants is largely dependent on the plant species, form of phytoestrogen ingested, mode of administration and adaptation of the rumen microbes to degrade the ingested metabolites. Hence, it has only recently been noted that lucerne exhibits estrogenic potency, as it produces lower concentrations of phytoestrogens compared to soybeans and some clover species. In ruminants, phytoestrogens are microbially degraded to non-estrogenic phenols [36] or passively absorbed into the bloodstream. Microbes in the rumen, once having been adapted to the presence of phytoestrogens, a process that may take approximately 7–12 days, will convert isoflavones such as genistein and biochanin A into non-estrogenic phenols (genistein into *para*-ethyl-phenol) [36,37]. From this, it may be likely that the cows in the L7 treatment may have had sufficient time for the rumen to adapt to the presence of phytoestrogens before the increase in plant phytoestrogen concentrations on day 28, compared to the L4 treatment group. However, the phytoestrogens formononetin and daidzein are demethylated into equol [14]. The metabolites produced from these phytoestrogens have greater estrogenic activity than their precursors and increase prostaglandin synthesis [37]. While equol was not detected in the plasma samples, we cannot rule out its potential presence in other samples.

Unlike isoflavones, coumestans are less likely to undergo degradation by rumen microbes, making coumestans such as coumestrol one of the more potent phytoestrogens [14]. Although the metabolism of individual animals [38] and species may vary for the phytoestrogens ingested, in non-ruminant species, isoflavones are less likely to undergo microbial degradation. The concentrations of formononetin remained above 5 mg/kg of DM for the first 42 days of the grazing trial in the GC7 treatment. The concentrations of formononetin can be attributed to the 19% of clovers present in GC7 (Appendix B, Table A1). This was similarly reflected in the plasma of the GC7 treatment; however, the concentrations of biochanin A were greater in plasma compared to the pasture. Formononetin and biochanin A have previously been observed to reduce conception rates [39,40] by reducing progesterone (P4) concentrations [41]. However the concentrations of biochanin A and formononetin present in the pasture of the current study were not as high as those reported by Hashem et al. (2018) [41]. It is unclear why the GC7 group did not produce any embryos; a range of factors may have contributed including pasture nutrition, the age of the cow and the stage of the cycle.

### 4.2. Effects on the Hypothalamo–Pituitary–Gonadal Axis

In the early 1960s, it was established that there were important differences in estrogenic action and the concentration of individual isoflavones not only between clover species but also between different cultivars of clovers [42]. Additionally, the negative effects observed in the reproductive tracts of females have been noted to be highly dependent on estrogenic action, the age and duration of exposure, the timing of exposure and the route of administration [4]. From the results observed in this study, the effects of phytoestrogens are more directly correlated to the timing of exposure with respect to concentrations in the pasture, rather than the length of exposure. A study on the binding of phytoestrogens to the estrogen receptors by Mathieson and Kitts [43] revealed lower progesterone concentrations in the CL tissues of heifers fed soybeans, compared to animals fed a standard feed mixture. The plasma P4 concentrations were highest in the G7 group, and lowest in the L4 treatment, suggesting that one of the potential reasons for reduced embryo recovery in the L4 group may be early embryonic loss, due to concentrations of phytoestrogens and insufficient P4. P4 concentrations were the lowest in the L4 group on days 14 and 42 (<0.05 ng/mL) (Figure A2). Progesterone concentrations were also low in the GC7 treatment on day 42 (0.08 ng/mL). The CL is essential for producing P4 to establish and maintain pregnancy; henceforth, the inhibition of the P4 production by phytoestrogens and their metabolites may result in a variety of negative impacts including the disruption of CL function and early embryonic mortality [44].

### 4.3. Factors Affecting Embryo Quality, Embryo Stage and Development

Embryo quality is of utmost importance in the survival of the embryo during early pregnancy and for undergoing embryo transfer programs. Embryo mortality can be a result of high levels of dietary protein or nitrogen during the time of conception, and it has been suggested that ruminants should avoid grazing on high protein/nitrogen pastures (such as lucerne) during the mating period [45]. While numerous phytoestrogens have been associated with disruption, it is known that in vitro genistein is able to block oocyte growth and disrupt follicle morphology [19]. Genistein and coumestrol, at 100 µg/mL and 10 µg/mL, respectively, have been observed to increase the number of immature oocytes in cows. From the concentrations recovered, 0.16–0.6 ng/mL of coumestrol and 0.02–0.14 ng/mL of genistein in plasma, the concentrations were not significant to affect the embryo quality at 7 days post AI. Whilst embryos were assessed using a non-objective technician, the IETS grading system is a subjective method of grading embryos unlike more advanced methods such as fluorescent staining techniques. A study conducted by Shore et al. 1998 determined that early embryonic loss was higher in cows that grazed on legume compared to a non-legume group [21]. The plasma concentrations observed at 42 days (two days post AI) in the L4 group were 0.42 ng/mL, higher than those recorded by Shore et al. 1998. This finding suggests that high concentrations of phytoestrogens may have influenced embryo recovery, 7 days post AI. Improved knowledge of phytoestrogen metabolism in ruminants is suggested because a high intake of key metabolites may result in adverse health effects, particularly in critical stages of the development of the conceptus and during lactation and pregnancy. Therefore, timing the exposure to phytoestrogens is key for capitalizing on health benefits while minimizing undesirable health outcomes [46].

### 4.4. Superovulation/Factors Affecting the Reproductive Cycle

An increase in multi-oocyte follicles has been observed as a clinical observation of phytoestrogens in mice [19]. The majority of the research on superovulatory programs as a correction for negative effects caused by high dietary phytoestrogen concentrations was conducted in mice. This study is one of a few investigating the effects of phytoestrogens on a bovine superovulatory program. Preliminary findings by Rodriguez et al. (1994) [47] suggested that phytoestrogens did not affect the superovulatory response in beef cows; however, the concentrations of phytoestrogens were lower than recorded in the current study. In the current study, the number of antral follicles observed was greater than the number of CLs recorded. The antral follicles in the G7 and L4 treatment were most likely to be immature, whilst within the L7 and GC7 treatments, these follicles may not have ovulated as a result of the phytoestrogens. The recovery of 56 embryos from 20 cows was a low recovery of embryos from a superovulation program.

## 5. Conclusions

The present study characterized the effect of phytoestrogens from lucerne, particularly coumestrol and formononetin, and their impact on the superovulatory response, embryo numbers, stage and quality. Our results suggest that increased concentrations of key phytoestrogens may affect the embryo stage, reduce P4 production and potentially contribute to early embryonic loss. Reproductive management regarding the grazing of lucerne and other legume pastures and managing the potential negative effects of phytoestrogens is important, as these legumes are commonly encountered in both cattle and sheep production in Australia and more globally. From our experimental field assessment, we suggest that grazing lucerne containing phytoestrogens at >25 mg/kg DM less than 20 days before artificial insemination may affect oocyte developmental competence, negatively impact embryo stage and reduce progesterone production, thus contributing to early embryonic loss. This was the first study to evaluate the concentrations of phytoestrogens in naturalized Australian clovers (haresfoot and cluster), demonstrating that these clovers too have the potential to adversely influence fertility in grazing livestock. A further characterization of this syndrome in sheep, horses and dairy cows could contribute valuable information to those involved in related livestock industries. Our findings suggest that grazing lucerne with low concentrations of phytoestrogens, for an extended period of time, may also reduce the risk of negative consequences on reproduction as the rumen may have reasonable time to adjust to the presence of bioactive phytoestrogens.

## Figures and Tables

**Figure 1 animals-14-01125-f001:**
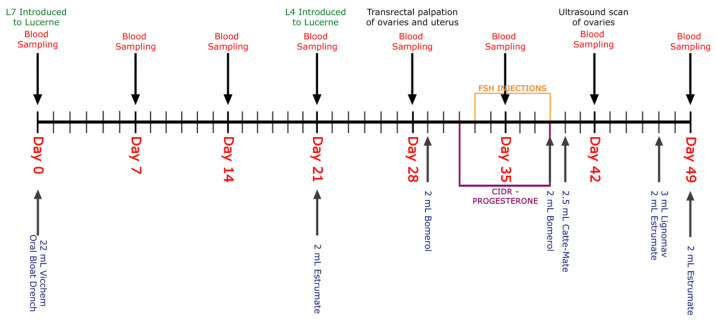
Summary of different protocols/procedures during experimental period of 7 weeks.

**Figure 2 animals-14-01125-f002:**
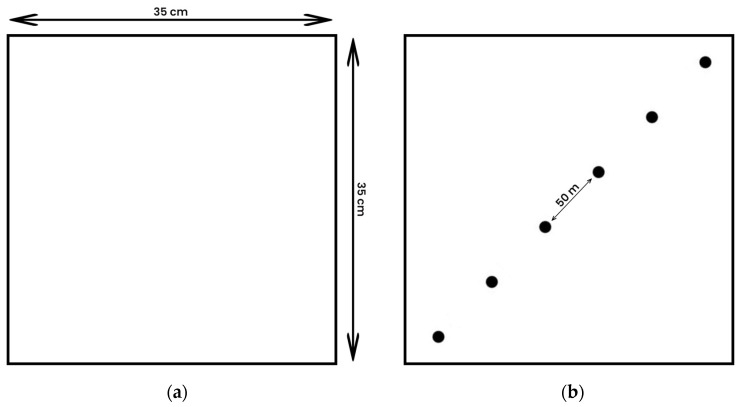
Visual assessment of pasture sampling method, (**a**) quadrat (35 × 35 cm^2^) used to collect pasture sample and assess pasture height, (**b**) quadrats on diagonal transect across pasture treatments.

**Figure 3 animals-14-01125-f003:**
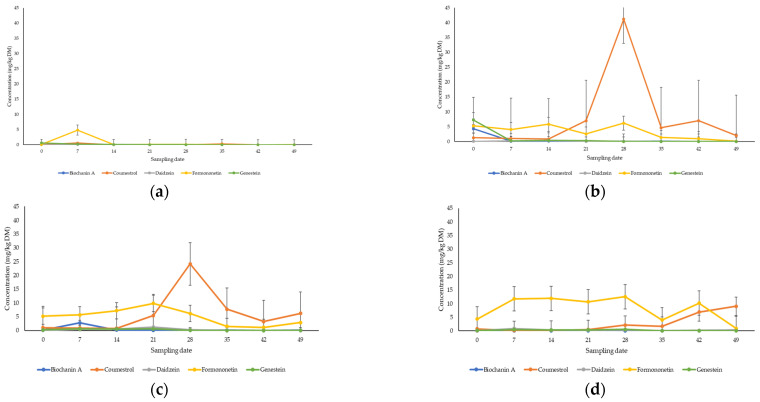
The average concentration of biochanin A, coumestrol, daidzein, formononetin and genistein in the fresh tissue extracts from the (**a**) G7—90% ryegrass, (**b**) L4—70% legumes (lucerne and arrowleaf clover), (**c**) L7—70% legumes (lucerne and arrowleaf clover) and (**d**) GC7—20% legumes (arrowleaf clover and naturalized clover)/75% ryegrass, over eight sampling dates.

**Figure 4 animals-14-01125-f004:**
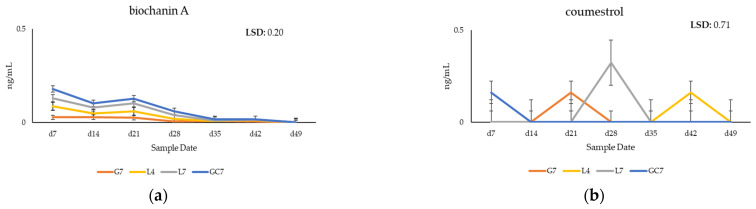
The average accumulated concentration of (**a**) biochanin A, (**b**) coumestrol and (**c**) daidzein and (**d**) formononetin and (**e**) genistein in the plasma (ng/mL) of the treatment and control groups over seven sampling periods with seven-day intervals.

**Figure 5 animals-14-01125-f005:**
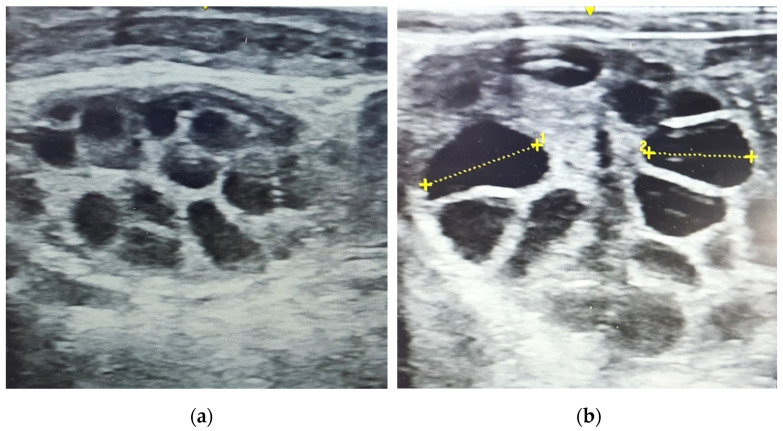
Ultrasound images of cow ovaries with multiple follicles (**a**) <1 cm and (**b**) >1 cm (^1^ 1.79 cm, ^2^ 1.51 cm).

**Figure 6 animals-14-01125-f006:**
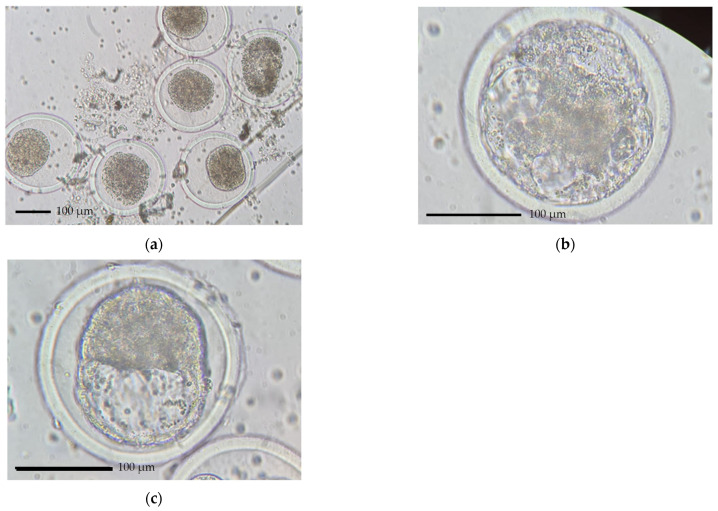
Light microscope images of the unfertilized oocytes and embryos recovered 7 days post AI showing three stages of embryo development in the treatment cows: (**a**) stage 1—unfertilized oocytes (UFOs) × 100 magnification, (**b**) stage 4—compact morula, ×400 magnification and (**c**) stage 5—early blastocyst, ×400 magnification. Scale bar, 100 µm.

**Table 1 animals-14-01125-t001:** The botanical composition and area of the control and pasture treatments in the grazing trial.

	Area	Lucerne %	Other Legumes ^a^ %	Desirable Grasses %	Broadleaf Weeds %
G7	5.07 ha	0	2	81	2
L7	4.73 ha	76	5	15	4
L4	4.73 ha	71	5	21	3
GC7	5.86 ha	0	19	73	2

^a^ Other legumes present in the botanical composition included the following: arrowleaf clover (*Trifolium vesiculosum*), white clover (*T. repens*), haresfoot clover (*T. arvense*), cluster clover (*Trifolium glomeratum*) and barrel medic (*Medicago truncatula*).

**Table 2 animals-14-01125-t002:** The baseline concentration (ug/mL) of phytoestrogens for the treatment groups in the grazing trial.

	G7	L7	L4	GC7
biochanin A	1.13	0.68	1.04	0.23
coumestrol	2.66	2.62	0.62	0.00
daidzein	14.02	20.81	1.02	0.00
formononetin	11.50	4.72	3.28	0.86
genistein	6.12	3.69	1.68	0.28

**Table 3 animals-14-01125-t003:** Phytoestrogenic metabolites from pasture legume and grass species identified by UPLC—QToF-MS in negative ionization mode.

Compound	Formula	Exact Mass g/mol	Basis for Identification ^1^	RT
3′methoxycoumestrol	C_16_H_10_O_6_	298.047738	AM	11.96
4′methoxycoumestrol	C_15_H_8_O_6_	282.052823	AM	11.4
apigenin	C_15_H_10_O_5_	270.052823	AM	9.45
apigetrin (apigenin 7-*O*-glucoside)	C_21_H_20_O_10_	432.105647	AM	6.7
azelaic acid	C_9_H_16_O_4_	188.104859	AM	7.45
biochanin A	C_16_H_12_O_5_	284.068474	STD	11.26
coumestrin (coumestrol 3-*O*-glucoside)	C_21_H_18_O_10_	430.089997	AM	7.6
coumestrol	C_15_H_8_O_5_	268.037173	STD	9.53
daidzein	C_15_H_10_O_4_	254.057909	STD	8.3
ferulic acid	C_10_H_10_O_4_	194.057909	AM	6.7
formononetin	C_16_H_12_O_4_	268.073559	STD	10.0
genistein	C_15_H_10_O_5_	270.052824	STD	9.37
hyperoside (quercetin 3-galactoside)	C_21_H_20_O_12_	464.095476	AM	6.8
isorhamnetin 3-*O*-glucoside	C_22_H_22_O_12_	478.111126	AM	7.2
luteoloside (luteolin 7-glucoside)	C_21_H_20_O_11_	448.100561	AM	7.3
*p*-coumaric acid	C_9_H_8_O_3_	164.047344	AM	6.3
pratensein	C_16_H_12_O_6_	300.063388	AM	10.6
quercetin	C_15_H_8_O_6_	302.042653	AM	8.7
rutin	C_27_H_30_O_16_	610.153385	AM	6.6

^1^ Basis for identification codes: AM—match to accurate mass/molecular formula; STD—match to accurate mass and retention time (RT) of analytical standards.

**Table 4 animals-14-01125-t004:** The number of follicles (<1 cm and >1 cm) and antral follicles (AF) present on both the right ovary (a) and the left ovary (b) on day 35 and day 39 of the protocol for the control and the treatment cows.

	Day 35			Day 39		
	>1 cm	<1 cm	AF	>1 cm	<1 cm	AF
G7	5	1	41	14	2	75
L4	4	0	40	14	2	76
L7	5	0	67	17	2	66
GC7	4	2	45	13	5	75

**Table 5 animals-14-01125-t005:** Number and grade of embryos recovered from various cow management treatments.

	Grade 1	Grade 2	Grade 3	Grade 4
G7	13	3	7	0
L4	9	0	0	0
L7	10	10	4	0
GC7	0	0	0	0
TOTAL	32	13	11	0

## Data Availability

All data are stored in archived datasets as per the guidelines of Charles Sturt University and associated funding bodies.

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
