# Peer review of "Effects of Mixed Pasture Legume Phytoestrogens on Superovulatory Response and Embryo Quality in Angus Cows"

_animals, 2024, doi:10.3390/ani14071125_

Round 1

Reviewer 1 Report

Comments and Suggestions for Authors

ANIMALS MS, March 2024

This is a most welcome contribution to industry where anecdotal accounts of fertility impairment, associated with lucerne, are occasionally aired.  The conclusion should be a useful alert and hopefully encourage further research into the challenge of balancing the feeding value of lucerne with the potential limitation to cows/ewes/horses fertility from coumestans.

My comments focus on the experiment’s description and the treatment pastures. The description of both lack standard details required by researchers seeking to reproduce the study.  I trust the description can be suitably expanded.

TITLE   Could the term “legume” be more specific?

L 26. “28 and 49 days”. In the text you mainly refer to 4 and 7 weeks. Consistent use of terms will improve readability. There are many acronyms to grapple with.

L 50-52.  This sentence may reflect a claim from Ref 7. A ref. to work that included relevant comparisons would be pertinent. The superior feeding value of legumes relative to grasses deserves mention. Your emphasis on protein, minerals and low fibre is appreciated; mention of different voluntary intake values should also be considered. 

L 53.  “drainage” ?:  would utilization of sub soil moisture be more accurate?

L 70-72.  Do not the effects of the biotic stresses you refer to apply mostly to the spp. that produce coumestans rather than isoflavones? If so, be specific: Medicago spp.

L 128.  “ryegrass” :  perennial/Italian/annual?  Please include scientific name(s).

L 132.  Detailed description is critical: viz. area of plots/paddocks. “4-replicated” suggests 4 plots of each pasture, but that is unclear.  The areas of the plots of 4 different pastures is not given. The measured amount of pasture may not limit the intake of five cows but unless the area of such pasture is known, we cannot assess that. Positive LWG over the 7 weeks could be reassuring, were it known.

How many quadrants?

Table 1:  Describe the 4 treatments within the column rather than L, R etc. The treatment terms are confusing. “Control” was also a 7 week treatment but that is not indicated in the term as it is for the others.   While the treatments botany needs to be fully described in the text, would the short descriptors be neater/more helpful if termed as G7, GC7, L4, L7? G = grass, C = clover?

The proportion of the various clover spp. should be indicated with some reference as to their known phytoestrogen characteristics of each.

The details of the experimental pastures are critically important and require detail: -

L150.  “Genesis”:  it would be desirable to describe the resistance to foliar diseases that are claimed for this cultivar by its owner. The age of the lucerne stand and the cultivar(s) of sub and arrowleaf  clovers should be identified.

L 215.  “6 reps.”: What was replicated 6 times? Do you mean 6 quadrants were measured in each plot? Why were 4 sub samples needed? How were they then used? Were they averaged?

You state that pasture samples were collected and analysed. That is to suggest that the phytoestrogen data apply to the pasture and not the lucerne or clover fraction per se. That should be made quite clear as the results may underestimate the result expected in a more dominant stand of lucerne.

For details of pasture measurement you refer us to the Wyse et al 2021 paper. That states that “Pasture samples collected from the treatment and control paddocks were sampled using a one-meter quadrant, with five sub-samples collected from each quadrant (modified method from McIntyre [40]).”

This still does not provide the information one needs to reproduce your sampling procedure. Was the quadrant subdivided and five subdivisions harvested based on ranked sets sampling. I do not follow how pasture height was used to calculate pasture mass.

I was surprised to see that George A McIntyre was publishing in 2005 - 31 years after his decease. I cannot access that paper but his oft-cited paper, “A Method for Unbiased Selective Sampling Using Ranked Sets,” was published in 1952 in the Australian Journal of Agricultural Research, 3, 385-390. http://dx.doi.org/10.1071/AR9520385. Would it be better to cite his AJAR paper?

The method used to quantify pasture is unclear and seems peculiar: -

L221.  “Pasture sample measurement number”:  I do not know what that means?

“Pasture density”?:  this term usually refers to the no. of plants/unit area. Do you mean mass/area? The yield of available pasture is readily measured by cutting and drying representative samples.

Fig 2.  The X axis needs explaining.  Is “frequency” referring to day no., the term used in Fig 1? Fig 3 uses “sample date”.  Is there an opportunity to use terms more consistently?

L 385-390.  Can LSDs be shown? Five cows per (unreplicated) treatment seems an extremely small number if differences were to be significant?

Fig 4.  In the caption you refer to “four grade stages”. Meaning? At L 204-6, you describe 8 stages and 4 grades.

L 411.  Should such “recommended levels” be given in the Introduction – with relevant references?

L 39.  “aphid” ?:  provide scientific name(s).

L 484.  In view of the important points you explain in section 4.2, would it be useful to describe the pasture/feed the cattle were provided with in the pre-experimental period?

L494. Check English.

I trust my comments can help make the paper easier to understand. We want to know exactly what was done.

I hope you can continue this field of investigation. 

Reviewer 2 Report

Comments and Suggestions for Authors

In the present work, Wyse et al. try to explain the effects of legume phytoestrogens on superovulatory response and embryo quality in Angus beef cattle. The present work suggest that concentrations of > 25 mg/kg DM of phytoestrogens less than 20 days preceding artificial insemination have negative effects on embryo development, progesterone production, which lead to production of immature oocytes and early embryonic loss. However, some questions also should be explained.

1. English grammar and writing style should be checked throughout the manuscript, for example,

Line 15, ‘Cattle were grazed on legume’, ‘cattle’ should be changed to ‘cows’. Please check this throughout the manuscript.

Lines 21-24, ‘The aim of this trial was to determine if coumestrol and other key phytoestrogens influence the superovulatory response in Angus cattle grazing legume pastures with a focus on the quantitative and qualitative assessment of embryos in contrast to beef cattle on non-legume pasture (control).’ This sentence should be rewritten.

Line 91, ‘PG E2’ should be changed to ‘PGE2’.

Lines 387, 397, ‘(F(9,64)=1.08’, ‘(F(7,21)=2.34’, which are right?

2. The Introduction section is too long, and it should be rewritten.

3. Materials and Methods section

Line 137, ‘Table 1’ is not in the correct place.

 ‘CIDR®’ or ‘CIDR’

Lines 137 and 214, ‘35 x 35 cm2 quadrants’ and ‘35 x 35 cm quadrats’, which are repeated?

4. Results section

Figure 4, Bar should be added.

In addition, some supported pictures for this study may be needed. For example, ovaries were scanned for the number of follicles and size of follicles, which were scanned using what, and the pictures of the follicles.

5. Conclusions section

This section should be rewritten based on your data and previous studies.

Lines 407-418, this paragraph may be not necessary.

Subtitles for Conclusions section may be not necessary.

6. Conclusions section

This section should be refined.

7. There is a related reference that is not cited in this paper.

Gonzalez-Martin R, Palomar A, Quiñonero A, Pellicer N, Zuckerman C, Whitehead C, Scott RT Jr, Dominguez F. Phytoestrogens Present in Follicular Fluid and Urine Are Positively Associated with IVF Outcomes following Single Euploid Embryo Transfer. Int J Mol Sci. 2023;24(13):10852.

Comments on the Quality of English Language

Extensive editing of English language required.

Round 2

Reviewer 2 Report

Comments and Suggestions for Authors

Thanks for author’s responses. However, Bars should be added in Figure 6. Line 772, delete ‘also’.

Comments on the Quality of English Language

Moderate editing of English language required.
